# All-Water-Driven High-k HfO_2_ Gate Dielectrics and Applications in Thin Film Transistors

**DOI:** 10.3390/nano13040694

**Published:** 2023-02-10

**Authors:** Fakhari Alam, Gang He, Jin Yan, Wenhao Wang

**Affiliations:** School of Materials Science and Engineering, Anhui University, Hefei 230601, China

**Keywords:** high-k, sol-gel, thin-film transistors (TFTs), water-driven (WD) routine, HfO_2_ thin films

## Abstract

In this article, we used a simple, non-toxic, environmentally friendly, water-driven route to fabricate the gate dielectric on the Si substrate and successfully integrate the In_2_O_3_/HfO_2_ thin film transistor (TFT). All the electrical properties of In_2_O_3_ based on HfO_2_ were systematically analyzed. The In_2_O_3_/HfO_2_ device exhibits the best electrical performance at an optimized annealing temperature of 500 °C, including a high µ_FE_ of 9 cm^2^ V^−1^ s^−1^, a high I_ON_/I_OFF_ of 10^5^, a low threshold voltage of 1.1 V, and a low sub-threshold of 0.31 V dec^−1^. Finally, test the stability of the bias under positive bias stress (PBS) and negative bias stress (NBS) with threshold shifts (V_TH_) of 0.35 and 0.13 V while these optimized properties are achieved at a small operating voltage of 2 V. All experimental results demonstrate the potential application of aqueous solution technology for future low-cost, energy-efficient, large-scale, and high-performance electronics.

## 1. Introduction

Metal oxide semi-conductors TFTs have been extensively researched for application in large-size active matrix flat-panels display, due to their outstanding high carrier mobility, optical transparency, and electrical stability. Based on metal oxide semi-conductors, TFTs are considered an improved substitute for the next generation of flat-panel display devices. The majority of work based on metal oxides up to date has been done for producing high-performance TFTs. 

As a significant switching control element, TFTs display devices have received a great deal of attention for their performance. Encouraged by the outstanding work of Teacher Nomura described in *Nature*, TFTs with different architectures have been built and applied to numerous fields to meet the demands of growing technological progress. Unfortunately, these TFTs devices are usually manufactured using expensive methods, such as atomic layer deposition, vacuum-based processing, sputtering, and pulse layer deposition. Offsetting its advantages, complex preparation processes and high manufacturing costs have become a major challenge for constructing high-performance, large-area electronic devices [1,2,3,4,5,6,7].

To address the issues as mentioned above, incredible developments in TFTs devices with a superior performance based on solution processing have been vigorously studied due to their small cost, large-area deposition capability, equipment simplicity, and simple mechanism of the composition ratio of every precursor. These enhancements have involved straightforward, balanced capability and versatility [8,9,10,11]. TFTs based on flexible substrates have attracted increasing attention owing to their exceptional advantages of flexibility, extensibility, and ultra-light weight. But the traditional solution-processed TFTs have good performance at high temperatures, which limits the use of flexible substrates [12,13,14]. Among these approaches, some harmless organic solvents, such as methoxyisopropanol and 2-methoxy ethanol (2-ME), are widely used to prepare precursor solutions. Undoubtedly, additional chemicals and subsequent production measures will increase manufacturing costs and environmental damage. Therefore, for high-performance TFTs, it is very important to use suitable gate dielectrics and environmentally friendly metal precursors at low temperatures [15]. 

The display industry of the future needs to be a pioneer in the solution-based process. As a result, it has been proposed to manufacture oxide TFTs in a water environment at low processing temperatures. Therefore, the water-based method is considered more environmentally friendly than traditional organic solvents. Compared with traditional organic solvents, water-based solutions are harmless and not toxic to the environment or the human body. Therefore, the water-induced solution process is a typical way of manufacturing oxide TFTs. In the latest reports, an environmentally friendly water encouragement method is used to prepare the channel and dielectric layer of an all-water-based oxide TFTs [16,17,18].

Similarly, the active layer and gate dielectric also play a vital role in describing the electrical performances of the TFTs devices. With the upgrading of microelectronics technology, low-voltage working TFT has become a typical way to obtain low-cost, low-power electronic devices. Generally, SiO_2_-based films require high voltage (30–40 V) to operate. In fabrication, the TFTs’ input power or operating voltage can be reduced, per the principle of field effect transistors, by utilizing a large-capacitance dielectric [19]. According to the research of Lee et al., the versatility of field-effect mobility can be improved by increasing the dielectric gate capacitance, which is attributed to the gate insulator’s higher dielectric constant than SiO_2_. Successively, extensive research was conducted on high-K materials, such as AlO_x_, ScO_x_, and HfO_x_; due to their exceptional thermal stability, high band gap and medium to high-k value, Hf-based dielectric materials have been given more attention [20,21]. The Hf-based dielectric materials have been given more attention, due to their high band gap and their exceptional thermal stability and they have a larger dielectric constant. So far, HfO_2_ TFT has been effectively implemented through solution-based processes. Although traditional methods rely more on organic solvents, they are separated during the formation of the HfO_2_ film, resulting in uncontrollable currents and nanopores [22]. In contrast, HfO_2_ film would be a better choice for preparing water-based coatings and would solve the above problems. As a possible candidate material for the active layer in transparent electronic devices, indium oxide (In_2_O_3_) has attracted attention due to its optical transparency and high electron mobility in the observable area. Indium oxide shows excellent electrical properties, including insulating properties and metal semiconductors, but is prone to defects and stoichiometry problems in the material. Compared with the advantages of the sol-gel method of indium oxide (In_2_O_3_) materials, it can be observed that the solution method has been considered a strong applicant for the preparation of high-performance oxide thin-film transistor devices [16,23]. 

In the current work, we have demonstrated a completely non-toxic solution path technology, a modest and large-area compatible deposition method in which HfO_2_ and In_2_O_3_ can be used to make a dielectric layer and channel layer on the Si subject. The HfO_2_ dielectric and structural properties were examined in detail to determine the effect of different processing temperatures on the HfO_2_ dielectric. To investigate the prospects of In_2_O_3_ films as channel layers, their application in TFT devices based on HfO_2_ dielectrics was also verified. In detail, it has been confirmed that HfO_2_/In_2_O_3_ TFTs exhibit excellent performance at a small operating voltage of 2 V, with a high I_ON_/I_OFF_ of around 10^5^ and a high field effect mobility µ_FE_ of 9 cm^2^ V^−1^ s^−1^. Finally, test the bias stress stability with threshold voltage shifts (V_TH_) of 0.35 and 0.13 V under PBS and NBS for the 1800 s respectively. It is a significant step in the success of small power consumption, low cost, and large-area full-oxide flexible electronic products.

## 2. Experimental 

### 2.1. Materials

The precursor used to prepare the HfO_2_ film was prepared by using hafnium chloride (IV) (99.5% Aldrich Shanghai, China). The precursor for manufacturing the channel layer prepared by using indium nitrate hydrate (III) (99.9%, Aldrich Shanghai, China ) is indium oxide. The precursor can be used without further purification.

### 2.2. Preparation of Precursor Solutions

The (0.1 M) HfO_2_ and In_2_O_3_ precursor solution were prepared by dissolving the hafnium chloride (HfCl_4_) Indium nitrate hydrate In(NO_3_)_3_·H_2_O in 10 mL di-water. The precursor solution was stirred by a magnetic stirrer for 6 h at room temperature. Before spin coating, the solution was aged in a moisture buster cabinet for 24 h and filtered through a 0.22 um injection.

### 2.3. Film Deposition and Device Fabrication

A p-type silicon wafer having a resistivity of 0.02 Ω cm was washed with the renewed RCA method, ethanol, and di-water, and dried with nitrogen. All silicon substrates are positioned in a plasma cleaner to increase the hydrophobicity of the surface of the Si substrate. The HfO_2_ solution was filtered through a 0.22-mm injection and then rotated on the hydrophilic silicon substrate at a speed of 500 rpm for 6 s and 4000 rpm for 20 s. The HfO_2_ film was then placed on a hot plate at 180 °C for 10 min to remove all residual solvent and cool to room temperature. This process has to be repeated 2–4 times to achieve the preferred thickness. The preferred thickness of the HfO_2_ thin film is approximately 15 nm. Finally, all HfO_2_ films were annealed in air at a temperature of 450 °C to 550 °C for two hours.

The In_2_O_3_ precursor solution rotated on the annealed HfO_2_ thin film at speed of 500 rpm, 6 s and 3500 rpm, for 20 s, after which the laminated sample was annealed in air at 275 °C for one h. Lastly, aluminum (Al) source and drain electrodes are deposited by thermal evaporation through a shadow mask. All devices’ channel lengths and widths are 100 and 1000 mm, respectively. The complete schematic diagram of the In_2_O_3_ TFTs solution based on the HfO_2_ film is shown in Figure 1.

### 2.4. Characterizations

An X-ray diffractometer (XRD) was used to investigate the microstructure of HfO_2_ film. Observable ultraviolet spectroscopy (UV-Vis, Shimadzu, UV-2550) was performed to study the annealing temperature-dependent bandgap and transmittance spectra of HfO_2_ films. Using a thermogravimetric analyzer (TGA, Pyris 1), the thermal-gravimetric analysis of the HfO_2_ xerogel was checked at a heating rate of 10 °C/min under airborne environmental conditions. X-ray photoelectron spectroscopy (XPS, ESCALAB 250 Xi, Thermo Scientific) was used to analyze the chemical composition of HfO_2_ film. 

Atomic force microscopy (AFM, SPA-400) was used to study the surface morphologies. The electrical characteristics of Si/HfO_2_/In_2_O_3_ TFTs were calculated by using an Agilent B1500A semiconductor device analyzer in the dark.

## 3. Results and Discussion

### 3.1. Microstructure Analysis of HfO_2_ Thin Films

To study the microstructure of the HfO_2_ thin films, XRD measurements were performed. Figure 2 shows the effect of the annealing temperature. It can be observed that HfO_2_ thin films remain amorphous at a lower temperature. The amorphous nature of thin film becomes less apparent as the annealing temperature rises. The polycrystalline state of the HfO_2_ thin film has been identified as increasing the annealing temperature, which leads to the HfO_2_ phase being thermally activated. The tetragonal phase of HfO_2_ is attributed to six main peaks centering at 24.5, 28.3, 31.6, 36.0, 41.4, and 45.7, which correspond to the (110), (−111), (111), (−102), (102), and (202) planes of HfO_2_. In TFTs, due to their high thermal stability and low leakage current, the amorphous state of the dielectric layer is more advantageous than its crystalline nature. The grain boundary becomes a profitable way for leakage current and impurity diffusion, leading to the display of off-current TFTs devices. Apart from the typical amorphous dielectric structure, the planar surface morphology contributes to the best dielectric/channel heterogeneous interface. The smooth and sharp interface between the dielectric layer and the channel layer of a TFTs is very appropriate. As a result, it is possible to conclude that the annealing temperature in the gate dielectric application should be precisely organized [24].

### 3.2. X-ray Photoelectron Spectroscopy (XPS) Measurements for HfO_2_ Thin Films

To determine the bond configuration of O and Hf atoms of the solution-processed HfO_2_ films, the XPS characterization was conducted, as shown in Figure 3. In Figure 3a, the O 1s spectra is divided into two peaks at 530 and 531.1, respectively. The peak at 530 eV indicates the existence of Hf–O bonds, while the peak of 531.1 eV indicates surface oxygen-like Hf–OH bonds [25,26]. With the increase of the annealing temperature from 450 to 550 °C, the integral ratio of Hf–O bonds gradually increased from 54.8% to 66.6% indicating the formation of the most HfO_2_, while the integral ratio of Hf–OH bond decreased from 45.2% to 33.4%, which was due to the elimination of residual hydroxyl groups. Note that the existence of OH group can induce defective states in the dielectric layer., which increase the leakage current and decrease breakdown voltage. The amount of surface oxygen should be small enough to use an HfO_2_ thin film as a gate insulator [27]. There are two binding energy peaks for Hf 4f, Hf7/2 and Hf5/2, as shown in Figure 3b. Hf7/2 peaks occur at 17.05 eV, while the binding energy of Hf5/2 peak occurs at 18.74 eV. As the temperature increased from 450 to 550 °C, the binding energy strength ratio of Hf 4f5/2 to Hf 4f7/2 decreased from 1.55 to 0. 97 indicating a slight increase of the Hf 4f5/2 binding state. In Figure 3b both peaks are well in accordance with the standard reference value of HfO_2_ from NIST (National Institute of Standard and Technology, USA Gaithersburg MD) XPS database [28,29].

### 3.3. AFM Analysis of HfO_2_ Thin Films

Critically, an appropriate condition of the material used in the TFTs devices is that the gate dielectric must be smooth. The surface morphologies of HfOx thin films on Si substrates are shown in Figure 4. The surfaces of HfOx-450, HfOx-500, and HfOx-550 were smooth, with root-mean-square (RMS) roughness values of 0.84, 0.86, and 0.90 nm, respectively. The dielectric of a thin film with a smooth surface can overturn the formation of the interface traps, decrease the transporter scattering centers, and gain extraordinary field effect mobility, which is valuable for increasing the performance of TFTs devices. From Figure 4, it is concluded that when we increase the annealing temperature, roughness increases. The thickness of the annealed thin films was favorable for the roughness growth, which leads to the crystallite size, as confirmed by the XRD.

### 3.4. Thermogravimetric Analysis

The mechanism of the HfO_2_ film was examined by using thermogravimetric analysis heated at a speed of 10 °C/min in air. As Figure 5 shows, the initial loss in weight at 180 °C was related to the evaporation of the residual solvent. The continuous weight loss occurred at 550 °C, which is related to the decomposition and dehydroxylation of hafnium chloride. There is no substantial weight loss above 600 °C, which indicates that the xerogel is converted to hafnium oxide, because the hafnium chloride is highly hygroscopic. Since the cation M^4+^ has a strong attraction to O^2−^, when it is dissolved in water, it will instinctively react with the water. Through the hydrolysis reaction, the hydroxy chloride of HfOCl_2_.8H_2_O hafnium is produced. The rapid violent hydrolysis of hafnium chloride in water indicates the formation of HfOCl_2_.8H_2_O. According to the hydrolysis reaction, hydroxyl chlorides are produced when water molecules react with hafnium chloride, as shown in Equation (1) [25,26,27,28,29,30].
(1)$${\mathrm{HfCl}}_{4}+H_{2}O=\;\mathrm{Hf}\left(\mathrm{OH}\right){\mathrm{Cl}}_{3\;}+\mathrm{HCl},$$
(2)$$\mathrm{Hf}\left(\mathrm{OH}\right){\mathrm{Cl}}_{3}\to {\mathrm{HfOCl}}_{2}+\mathrm{HCl}.$$

After spin-coating the solution on the substrate, the deposition precursor film contains HfOCl_2_·nH_2_O. When HfOCl_2_ is thermally annealed, it decomposes and transforms into HfO_2_. The general reaction of HfOCl_2_ is described in Equation (3).
(3)$${\mathrm{HfOCl}}_{2}+{\mathrm{nH}}_{2}O\to {\mathrm{HfO}}_{2}+2\mathrm{HCl}+\left(n-1\right)H_{2}O$$

In the case of an aqueous solvent-based process, the as-deposited precursor film contains strongly hydrated HfOCl_2_. HfOCl_2_∙nH_2_O with n ≥ 4 has a tetramer structure. This structure consists of [Hf_4_(OH)_8_(H_2_O)_12_]^8+^ and eight chloride ions [31,32]. Every Hf atom shares a double OH bridge with each of its two Hf neighbors, whereas Hf atoms are bridged by Hf–O–Hf or Hf– OH–Hf bonds in the HfOCl_2_∙nH_2_O with n < 4. Upon heating, the reaction of Cl^−^ and OH^−^ with Hf atoms from the tetramer leads to the formation of Hf_2_O_3_Cl_2_∙3H_2_O, which gives in turn Hf_2_O_3_Cl_2_·H_2_O, which leads finally to the formation of HfO_2_ and to a loss of HCl.
Hf_2_O_3_Cl_2_.3H_2_O → Hf_2_O_3_Cl_2_·H2O + 2H_2_O
Hf_2_O_3_Cl_2_·H_2_O → 2HfO_2_ + 2HCl

### 3.5. U-V Analysis of HfO_2_ Thin Films

The optical transmittance and absorbance of the HfO_2_ thin film are shown in Figure 6a,b. The transmittance range of HfO_2_ film on the glass substrate is calculated under the condition that the wavelength fluctuates between 200 and 900 nm, as shown in Figure 6a.

Figure 6a shows that the light transmittance of all films in the visible light range has reached more than 90 per cent, which may be applied to transparent electronic devices. No color difference between the sample and bare glass was detected, showing good optical transparency. As the temperature rises, the transmittance drops slightly, which helps increase the surface roughness of the HfO_2_ film or remove oxygen defects at higher temperatures [33]. The optical bandgap of the HfO_2_ film was measured by using the standard Tauc plotting method, as shown in Figure 6b.

The bandgap of the HfO_2_ film is plotted against the photon energy (hυ)^2^. The calculated values of the band gap of the HfO_2_ film are 5.6 eV, 5.4 eV, and 5.2 eV, respectively, related to the annealing temperatures of 450 ^0^C, 500 °C, and 550 °C, respectively. It can be observed that the bandgap of the film decreases with the rise of the annealing temperature, which is due to the smaller particle size of the HfO_2_ film [34].

### 3.6. Areal Capacitance of WI HfO_2_ Thin Film

To explore the dielectric areal capacitance of WI HfO_2_ film induced by water treated at different annealing temperatures, MOS capacitors with an Al/HfO_2_/Si structure were prepared. Figure 7 demonstrates the areal capacitance (C) in the range of 20 Hz to 100 kHz related to frequency (f). The areal capacitance of HfO_2_ thin films was annealed at different temperatures of 450 °C, 500 °C, and 550 °C, which are measured to be 420, 520, and 1225 nF/cm^2^, respectively. It is evident from Figure 7 that areal capacitance increases as the temperature of thin films increases from 450 °C to 550 °C in the low-frequency region. The increase in the areal capacitance of thin films annealed at variable temperatures may result in the capacitance of metal hydroxide being less than that of metal oxide and the thermally enhanced dehydroxylation reaction [17,34].

### 3.7. Electrical Properties of Solution-Processed of In_2_O_3_/HfO_2_ TFTs

To study the efficiency of HfO_2_ thin film as a gate dielectric, an In_2_O_3_ TFT with a bottom-gate and top-contact structure based on an HfO_2_ dielectric was constructed. The HfO_2_ dielectric was annealed at various temperatures between 450 °C and 550 °C. Figure 8a–f shows the corresponding transfer and output curves of an In_2_O_3_ TFT based on a HfO_2_ dielectric. The electrical properties of thin film transistors are shown in Table 1. The high I_ON_/_IOFF_ current ratios of the In_2_O_3_ TFTs based on HfO_2_-450 °C, HfO_2_-500 °C, and HfO_2_-550 °C are 10^3^, 10^5^, and 10^4^, respectively. The field-effect mobility (μ_FE_) in the saturation region was calculated by using the following Equation (4).
I_DS_ = W/2L μ_FE_ C_i_(V_GS_ − V_TH_)^2^
(4)
where L and W are the mask’s channel length and width, and Ci is the areal capacitance of the HfO_2_ thin film. The field-effect mobility μ_FE_ for the In_2_O_3_ based on HfO_2_-450, HfO_2_-500 and HfO_2_-550 was calculated to be 5.4, 9 and 7.8 cm^2^ V^−1^ s^−1^, respectively.

The In_2_O_3_ TFT based on HfO_2_-500 displays the best electrical properties, such as a field-effect mobility of 9 cm^2^/V^−1^ s^−1^, a high I_ON_/I_OFF_ 10^5^, a small sub-threshold swing (SS) of 0.31 V dec-1, V_TH_ of 1.1, and a Dit of 1.2 × 10^13^. Some of the electrical properties of In_2_O_3_ TFTs found by using the solution process are shown in Table 2. Due to the large capacitance of the gate dielectric, The field effect mobility of the TFTs is very high. As we know, the mobility of thin film semiconductors depends on the transporter concentration, which is primarily defined by dielectric capacitance at the specified gate voltage in the indistinguishable device configuration. Export transportation is guided by osmotic conduction at the trap site, and when the carrier concentration is high, it will increase due to the abundance of trap sites.

These electrical properties were obtained at a low voltage of 2 V, which is important for applications in low-power electronic devices. The sub-threshold swing is defined as V_GS_ needing to increase I_DS_ by ten times. Generally, the sub-threshold swing principle directly reflects TFTs’ power consumption and switching speed. If we need to switch transistors quickly, we need a low sub-threshold swing rate. The sub-threshold swing value can be calculated from the transfer curve using Equation (5) [35].
(5)$$\mathrm{SS}\;=\frac{{\mathrm{dV}}_{\mathrm{GS}}}{{\mathrm{dlogI}}_{\mathrm{DS}}}$$

The SS values for In_2_O_3_ TFTs based on HfO_2_-450, HfO_2_-500, and HfO_2_-550 are calculated at 0.59, 0.31, and 0.46 V/dec, respectively. From Table 1, HfO_2_-500 shows us the smallest value of SS of In_2_O_3_ TFT, which is mostly useful from high areal capacitance and the smooth interface between In_2_O_3_ and HfO_2_ layers. Based on the sub-threshold swing value, the Dit interface density states can be calculated using Equation (6) [36].
(6)$$D_{\mathrm{it}}=\left[\frac{\mathrm{SS}\;\log\left(e\right)}{\mathrm{KT}/q}-1\right]\frac{C1}{q}$$
where k, T, and q are Boltzman’s constant, absolute temperature, and charge feature, respectively. The D_it_ values of the In_2_O_3_ TFTs annealed at 275 °C are calculated as in Table 1**.**

**Table 2 nanomaterials-13-00694-t002:** Field effect mobility of the solution process In_2_O_3_ TFTs based on various high-k dielectrics.

Temperature (°C)	Dielectric	Solvent	µ_FE_ [cm^2^ v^−1^s^−1^]	References
600	SrO_x_	2-ME	5.61	[37]
300	LiO_x_	2-ME	5.69	[38]
500	YbO_x_	2-Methoxyethanol+N, N-dimethylformamide	4.98	[39]
200	ZrO_2_:B	2-ME	4.01	[40]
500	MgO	2ME	5.48	[41]
350	SiO_2_	2ME	3.53	[42]
500	HfO_2_	H_2_O	9	Current work

The D_it_ value of In_2_O_3_ TFTs is reasonably acceptable related to the reported TFTs based on solution processes such as MgO and ScOx, which have 10^13^ cm^−2^ [17,43]. The typical output characteristics of In_2_O_3_ based on HfO_2_-450, HfO_2_-500, and HfO_2_-550 are shown in Figure 8d–f at various gate voltages ranging from 0 to 2 V in each step of 0.5 V. The device displays typical n-type FET actions with clear pinch-off and saturation. These enhanced properties are assumed to result from using high-k HfO_2_ dielectric and the smooth interface between the HfO_2_ high-k and the In_2_O_3_ active layer. Furthermore, the brilliant electrical properties of In_2_O_3_ TFTs based on di-water-induced HfO_2_ demonstrate the potential for rapid progress in the transparent oxide electronics of solution at low construction costs for the next generation. Figure 9a,b shows us the stability of PBS and NBS -induced change in transfer curves of HfO_2_/In_2_O_3_ TFTs. The gate voltage of 2 V is constantly applied to the TFT at 1800 volts throughout the test. The transfer curves of HfO_2_/In_2_O_3_ TFTs shift to the right by 0.35 and 0.13 V with increasing bias application time, showing that the stability of HfO_2_/In_2_O_3_ TFT has been enhanced. The NBS of HfO_2_/In_2_O_3_ TFT was also examined, and the effects are shown in Figure 9b. In the NBS test, HfO_2_/In_2_O_3_ TFTs display the negative V_th_ drift. In the situation of the PBS, the highest intention of the threshold voltage shift shown in Figure 9a is that the electrons continue to take their place in the thin film associated with the electric field, and electron aggregation happens at the interface between the gate dielectric and thin film. Differences in the Fermi energy-trapped electrons, which are collected at the interfaces of thin film and bulk sites, cause oxygen vacancy neutralization (V_O_ + e = V_O_ and V_O_^2+^ + 2 e^−^→ V_O_) [44]. Therefore, the electron capture happens at the interface when the applied gate voltage is inhibited. At that point, the gate’s electric field vanishes, and this portion of the charge cannot be inhibited quickly into a semiconductor. As a result, an electric field is created at the interface between the high-K and channel layers, which has the opposite effect of the gate electric field. If you turn on the TFT device, a higher voltage must be applied to the gate to counteract certain effects of the electric field at the interface. As a result of the trap in the PBS, a positive shift takes place. On the other hand, electron discharge from donor-like traps and oxygen vacancies causes a negative shift in V_th_ (V_O_→2e^−^ + V_O_^+2^ and V_O_ → + e^−^ +V_O_), which is shown in Figure 9b. During the NBS study, the off-current is once increased followed by a decrease. It is due to the defects and interfacial trapping centers between the dielectric and channel layer. In addition, the ionization of the oxygen vacancy V_o_^+^ is decreased, and the defects and trapping center increased due to the increasing annealing temperature of the dielectric layer of the device, so that is the main reason that the NBS the off-current increased, followed by a decrease.

## 4. Conclusions

In summary, HfO_2_ thin films are prepared on Si substrates by water-driven solution processes and annealed at various temperatures, successfully integrating the Si/HfO_2_/In_2_O_3_ TFTs devices, which are annealed at various temperatures. Si/HfO_2_/In_2_O_3_ annealed at 500 °C exhibits the best electrical properties, including a high I_ON_/I_OFF_ of 10^5^, a field-effect mobility of 9 cm^2^ V^−1^ s^−1^, a threshold voltage of 1.1 V, and a low SS of 0.31 V dec^−1^, as well as excellent bias stress stability, with V_TH_ of 0.35 and 0.13 for 1800 s under the PBS and NBS. These optimized electrical parameters are achieved at a small operating voltage of 2 V. The transmittance of all HfO_2_ films in the visible light range exceeds 90%. HfO_2_ films annealed at different temperatures show good large-area capacitance characteristics, which is an important step towards achieving lower power consumption, low cost, energy efficient, large scale, and high performance of electronics devices.

## Figures and Tables

**Figure 1 nanomaterials-13-00694-f001:**
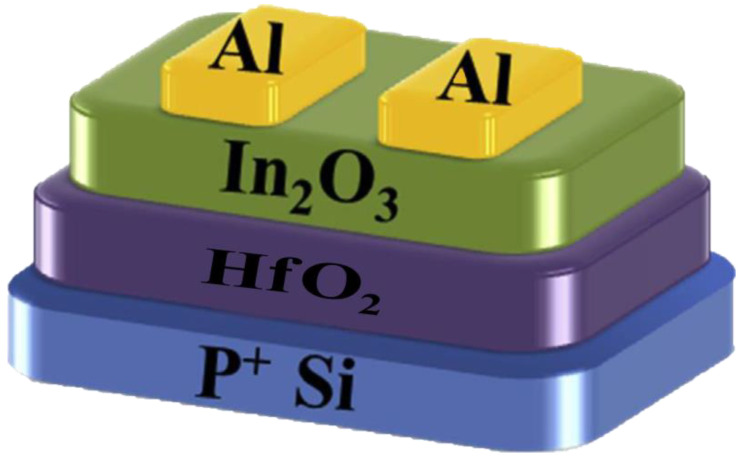
Schematic diagram of In_2_O_3_ TFT with HfO_2_ gate dielectric.

**Figure 2 nanomaterials-13-00694-f002:**
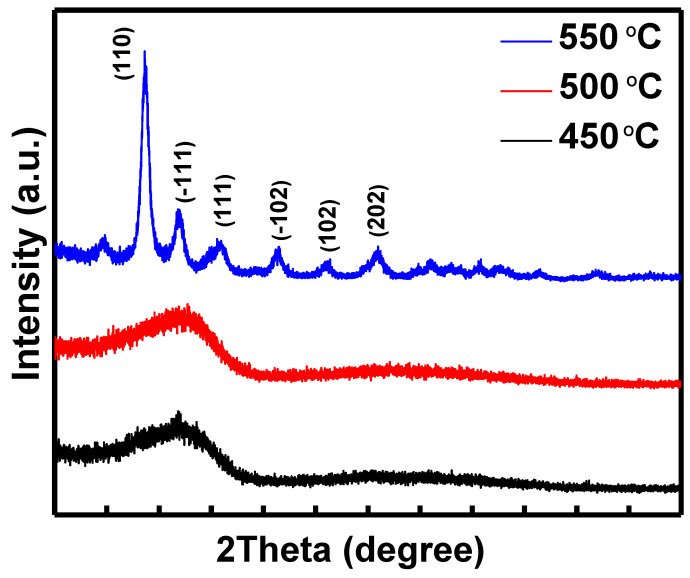
GIXRD patterns of the HfO_2_ films with various temperatures.

**Figure 3 nanomaterials-13-00694-f003:**
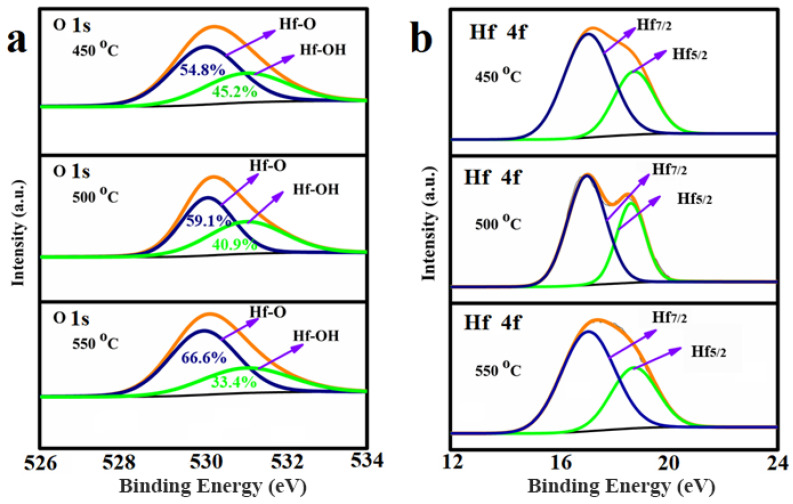
(**a**,**b**) XPS spectra of O 1s and Hf 4f peaks for HfO_2_ thin films as a function of annealing temperature.

**Figure 4 nanomaterials-13-00694-f004:**
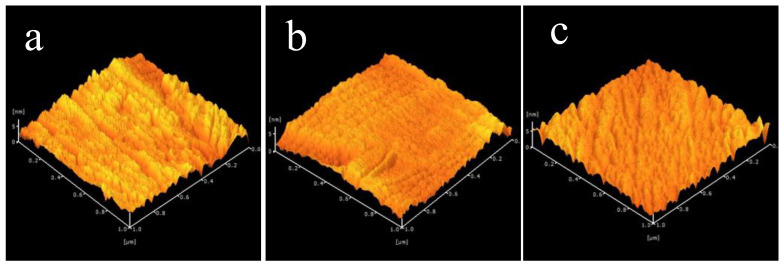
AFM images of HfO_2_ thin films with various Tempractures (**a**) 450 °C (**b**) 500 °C (**c**) 550 °C.

**Figure 5 nanomaterials-13-00694-f005:**
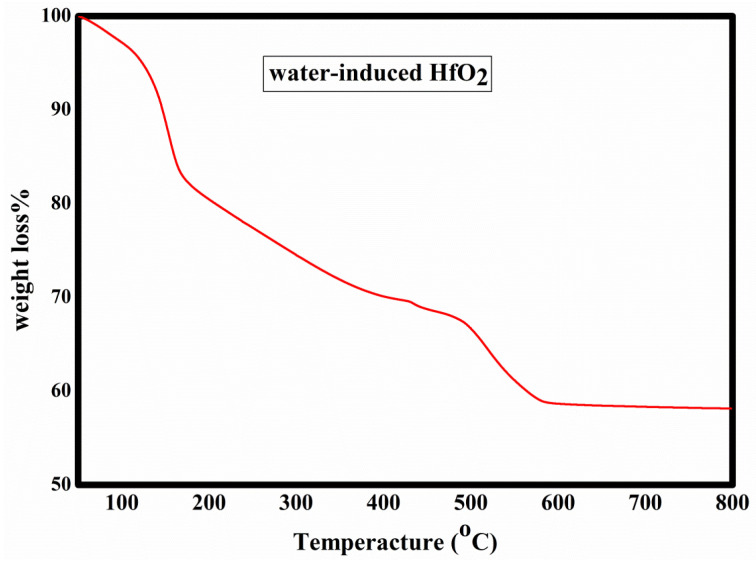
Thermal behavior of HfO_2_ xerogel.

**Figure 6 nanomaterials-13-00694-f006:**
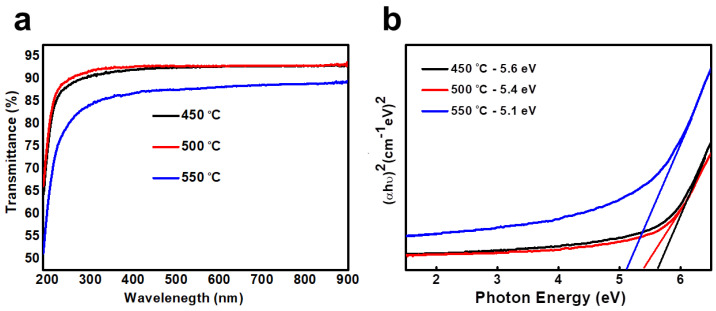
(**a**) Optical transmittances of WI HfO_2_ thin films annealed at different temperatures; (**b**) bandgap of WI HfO_2_ thin films annealed at different temperatures.

**Figure 7 nanomaterials-13-00694-f007:**
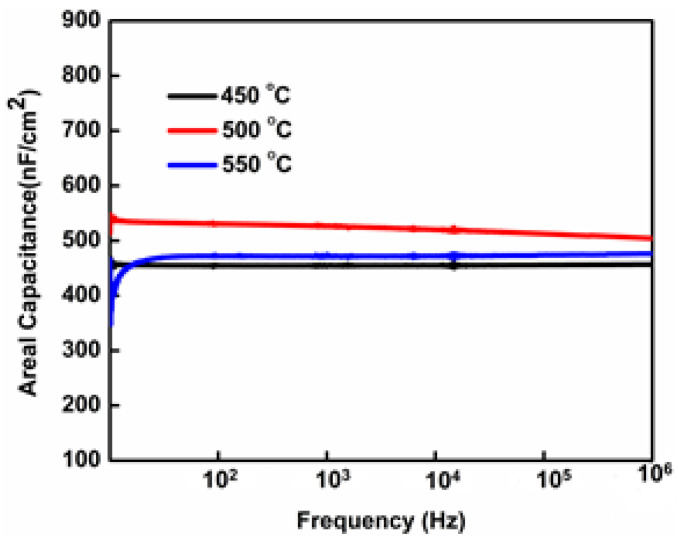
Areal capacitance of the WI HfO_2_; thin films annealed at different temperatures.

**Figure 8 nanomaterials-13-00694-f008:**
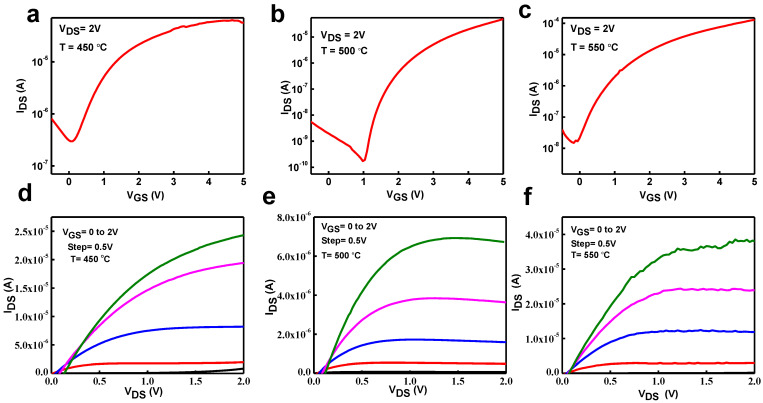
(**a**–**c**) Transfer characteristics of the In_2_O_3_/HfO_2_TFTs. (**d**–**f**) Output characteristics of the In_2_O_3_/HfO_2_TFTs.

**Figure 9 nanomaterials-13-00694-f009:**
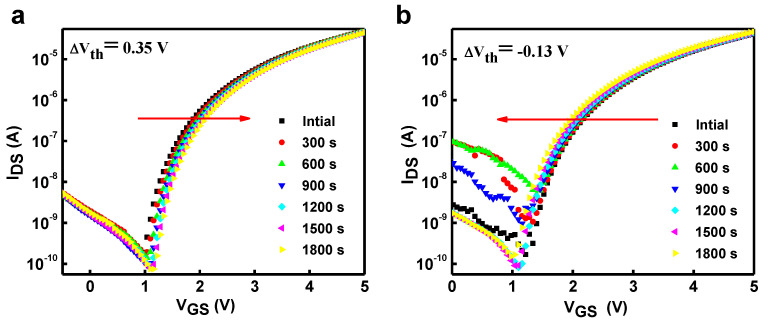
(**a**) PBS Transfer curve of the In_2_O_3_/HfO_2_TFTs; (**b**) NBS Transfer curve of the In_2_O_3_/HfO_2_TFTs.

**Table 1 nanomaterials-13-00694-t001:** Electrical properties of In_2_O_3_/HfO_2_ at different annealing temperatures.

Sample	Annealing Temperature	µ_FE_ [cm^2^ V^−1^ s^−1^]	I_ON_/I_OFF_	V_TH_ [V]	SS [V dec^−1^]	D_it_ [cm^−2^eV^−1^]
In_2_O_3_/HfO_2_	450 °C	5.4	10^3^	−0.06	0.59	3.7 × 10^13^
In_2_O_3_/HfO_2_	500 °C	9	10^5^	1.1	0.31	1.2 × 10^13^
In_2_O_3_/HfO_2_	550 °C	7.8	10^4^	0.6	0.46	2.8 × 10^13^

## Data Availability

Research data are not shared.

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
