# Peer review of "All-Water-Driven High-k HfO2 Gate Dielectrics and Applications in Thin Film Transistors"

_nanomaterials, 2023, doi:10.3390/nano13040694_

Round 1

Reviewer 1 Report

All-water-driven high-k HfO2 gate dielectrics and its application in thin film transistors

Judgement: Accept as is

Summary: In this work, the authors use a water-driven route to fabricate the gate dielectric and demonstrate In2O3/HfO2 TFTs with good electrical properties.

Comments: The authors should take into account the following comments before publication:

1.  The authors introduce the field of printable and flexible electronics as the target application, but end up annealing at 500C that is incompatible with flexible substrates. Can the authors provide some directions for improvement in future, such as UV annealing etc (eg ref https://doi.org/10.1021/acs.chemmater.6b03499)?

2.  The novelty of this work is not clear. Can the authors briefly describe the advantages and novelty of this work when compared to literature?

Reviewer 2 Report

The manuscript entitled “All-water-driven high-k HfO2 gate dielectrics and its application in thin film transistors” submitted by Fakhari Alam et al. reported on the preparation of water-based HfO2 gate dielectrics and the performances in In2O3 TFTs. The authors claim the simple, non-toxic, environmentally friendly, water-driven route of the reported procedure. The solution-processed In2O3/HfO2 TFTs exhibited a high mobility of 9 cm2/Vs, with an optimum HfO2 preparation condition.

Unfortunately, I do recommend the publication of this work in Nanomaterials at least in the current form because of a lack of reasonable experiments and presentation. The authors are required to address the following points.

1. The authors mention that the water-driven route shown in this work is non-toxic. However, as mentioned in chapter 3.4, HfCl4 produces toxic HCl when reacting with water. Thus I do not agree with the reviewers’ claim.

2. The thickness of HfO2 films cannot be found. As the authors mentioned “This process has to be repeated 2-4 times to achieve the preferred thickness.” at line 113, the each or final thickness of all annealing temperature conditions should be reported.

3. In chapter 3.2, XPS analysis is shown. The explanation should be provided in more detail. For instance, at line 162, the authors mentioned that "The intensity of the peak of 530 eV has increased with ..."; quantitative discussion is required. Besides, a peak intensity is not informative in XPS but an integration area is meaningful.

4. At line 171, the authors mentioned “As the annealing temperature rises from 450 °C to 550 °C, the peak intensity of Hf 4f5/2 increases while the peak intensity of Hf 4f7/2 decreases.” What does it mean? In principle, the 7/2:5/2 ratio is unique to be 4:3 due to the spin-orbit coupling, which is obviously not in the case of the present analysis (Figure 3b). Careful analysis and scientifically reasonable presentation must be done.

5. A minor point, the fitting lines in Figure 6b should be shown by another color to make it easy to distinguish them with the plot for 500 degC. It is also wondering if the legend is correct. The fitting line to the black curve (450 degC) seems to be extrapolated to 5.4 eV.

6. In Figure 7, 500dC shows a gradual decrease in capacitance even in the shown range, which is different from the others. Metal oxide dielectrics do not typically exhibit dielectric dispersion below 1 MHz or rather higher frequencies (as 450 and 550 degC samples herein). At least, the 500 degC film should be studied in more detail to verify dielectric properties of HfO2 obtained by the current procedure. Besides, the frequency (x-axis of this plot) should be shown in logarithmic scale and frequencies as high as possible.

7. Transfer curves in Figure 8a-c show high off-currents in all TFTs. As the increase in Ids by reverse Vgs is a unique feature of In2O3, I would consider that all TFTs show the gate leakage current probably comparable to Ids. Though, the leakage may not occur through defects in the HfO2 gate dielectrics but through the unpatterned I2O3 layer which electrically connects the source/drain and gate electrodes because the Ids curves do not pass through 0 A at Vds = 0 V in the output curves in Figure 8d-f. In either cases, the estimated field effect mobility values are not much reliable, thus Table 2 may not be meaningful. Also, the off-current shown in Figure 9 is lower than that in Figure 8. Please make it clear if the reported mobility values are an average or the maximum among several TFTs, and how reproducible the TFT performances are.

8. In Figure 9b, during the NBS study, the off-current is once increased followed by decreasing. The reason should be addressed

Reviewer 3 Report

This paper investigates the preparation and properties of thin-film transistors using hafnium oxide.. It is expected to contribute to the development of the field. It would be worthy of publication if the following points were corrected.

The authors use hafnium, but this element is not familiar to me as a transistor material. What is the reason for using hafnium instead of other universally used elements? In other words, what are the advantages of hafnium compared to other elements for this application?

The device structure shown in Fig. 1, what is the reason for stacking the layers in this order, and what would happen if the HfO2 and In2O3 layers were placed in reverse?

In the transfer curve in Fig. 9, Vth shifts over time, what changes are taking place in the device? Is it a degradation of some layer? Or are there irreversible changes between layers (e.g. delamination)?

It is well written. I consider it worthy of acceptance if the sections I commented on to the author can be corrected.

Round 2

Reviewer 2 Report

Unfortunately, I do not agree with the publication of the manuscript in the current form because they did not address my comments sufficiently, and there is a lack of scientific correctness.

1. Although I required quantitative discussion on O 1s XPS based on the peak integration area, it has not been improved.

2. As for Hf 4f XPS, my previous comment "the 7/2:5/2 ratio is unique to be 4:3 due to the spin-orbit coupling" means that a peak integral of 4f 5/2 cannot be larger than that of 4f 7/2 for the same chemical species. In addition, a binding energy of Hf 4f 7/2 is smaller than that of Hf 4f 5/2 by about 1.7 eV. Hence, the authors' assignment is definitely wrong.

3. Moreover, the authors discussion at lines 175-178 is scientifically nonsense. As mentioned above, 4f 7/2:5/2 ratio is unique, and hence, it is impossible that the peak intensity of Hf 4f5/2 increases while the peak intensity of Hf 4f7/2 decreases, as the authors mentioned.

4. Besides, there are some or many studies on Hf 4f XPS in hafnium oxides. They discuss HfO2 and HfOx formations with the scientific correctness.

Overall for the above regarding XPS, the authors must learn the fandamental and search the literatures before resubmitting the manuscript.

5. The authors corrected Figure 6b. However, nobody shall understand how and why the red straight lines are drawn as shown.

And an additional comment; it is highly encouradged to plot the source-gate current in Figure 8a-c to uncloud insularing properties of the dielectric layers.

Author Response

                                                           Response Letters

  1. Although I required quantitative discussion on O 1s XPS based on the peak integration area, it has not been improved.

Response: Thanks to the reviewer for the suggestion. Following the reviewer comment and suggestion, The O 1s XPS spectra had been divided into two peaks at 530 and 531.1 eV.  The peaks 530 eV indicates the existence of Hf–O bonds while the peak of 531.1 eV indicates surface oxygen like Hf–OH bonds [25,26]. With the increase of annealing temperature from 450 to 550 °C, the integral ratio of  Hf–O bonds gradually increase from 54.8% to 66.6% indicating the formation of the most HfO2 , while the integral ratio of Hf–OH bond decrease from 45.2% to 33.4% , which is due to the elimination of residual hydroxyl groups. Note that the existence of OH group can induce defects states in dielectric layer, which increase the leakage current and decrease breakdown voltage. the amount of surface oxygen should be small enough to use an HfO2 thin film as a gate insulator.

  1. As for Hf 4f XPS, my previous comment "the 7/2:5/2 ratio is unique to be 4:3 due to the spin-orbit coupling" means that a peak integral of 4f 5/2 cannot be larger than that of 4f 7/2 for the same chemical species. In addition, a binding energy of Hf 4f 7/2 is smaller than that of Hf 4f 5/2 by about 1.7 eV. Hence, the authors' assignment is definitely wrong.

Response: Thanks to the reviewer for the suggestion. Following the reviewer comment and suggestion, XPS graph had been corrected in the updated version of the mainscruipts. In previous version, on the XPS shown in Fig. 3b, I made a  mistake to write binding peaks of 4f 5/2 as 4f7/2 and vise versa. The numbering of the peak had   been corrected and explained as below. There are two binding energy peaks for Hf 4f, Hf7/2 and Hf5/2 as shown in figure 3b. Hf7/2 peaks centers at 17.05 eV and the binding energy of Hf5/2 peak occurs at 18.74 eV.  As the temperature increases from 450 to 550°C. the binding energy strength ratio of Hf 4f5/2 to Hf 4f7/2 decreases from 1.55 to 0. 97, indicating the slightly change of Hf binding state.

  1. Moreover, the authors discussion at lines 175-178 is scientifically nonsense. As mentioned above, 4f 7/2:5/2 ratio is unique, and hence, it is impossible that the peak intensity of Hf 4f5/2 increases while the peak intensity of Hf 4f7/2 decreases, as the authors mentioned.

Response :Thanks to the reviewer for the suggestion. Following the reviewer comment and suggestion the XPS graph had been corrected in the updated version of the manuscript.

  1. Besides, there are some or many studies on Hf 4f XPS in hafnium oxides. They discuss HfO2 and HfOx formations with the scientific correctness. Overall for the above regarding XPS, the authors must learn the fundamental and search the literatures before resubmitting the manuscript.

Thanks to the reviewer for the suggestion. Following the reviewer comment and suggestion, the XPS graph had been corrected and explained in the details as below. To determine the bond configuration of O and Hf atoms of the solution-processed HfO2 films, the XPS characterization had been conducted and shown in figure 3. In figure 3a, The O 1s spectra had been divided into two peaks at 530 and 531.1 respectively.  The peaks 530 eV indicates the existence of Hf–O bonds while the peak of 531.1 eV indicates surface oxygen like Hf–OH bonds [25,26]. With the increase of annealing temperature from 450 to 550 °C, the integral ratio of  Hf–O bonds gradually increase from 54.8% to 66.6% indicating the formation of the most HfO2 , while the integral ratio of Hf–OH bond decrease from 45.2% to 33.4% , which is due to the elimination of residual hydroxyl groups. Note that the existence of OH group can induce defects states in dielectric layer., which increase the leakage current and decrease breakdown voltage. the amount of surface oxygen should be small enough to use an HfO2 thin film as a gate insulator.[27] There are two binding energy peaks for Hf 4f, Hf7/2 and Hf5/2 as shown in figure 3b. Hf7/2 peaks occurs at 17.05 eV the binding energy of Hf5/2 peak occurs at 18.74 eV.  As  the temperature increases from 450 to 550°C. The binding energy strength  ratio of Hf 4f5/2 to Hf 4f7/2 decreases from 1.55 to 0. 97 indicating the slightly  increase of Hf 4f5/2 binding state. In figure 3b both the peaks which are well accordance with the standard reference value of HfO2 from NIST (National institute of Standard and Technology, USA) XPS database[28,29]

  1. Yoo, Y. B.; Park, J. H.; Lee, K. H.; Lee, H. W.; Song, K. M.; Lee, S. J.; Baik, H. K. Solution-processed high-k HfO2 gate dielectric processed under softening temperature of polymer substrates. J. Mater. Chem. C 2013, 1, 1651-1658.
  2. Shimizu, H.; Sato, T.; Konagai, S.; Ikeda, M.; Takahashi, T.; Nishide, T. Temperature-Programmed Desorption Analyses of Sol–Gel Deposited and Crystallized HfO2 Films. Japanese J. Appl.  Phys 2007, 46, 4209.
  3. Chua, L. L.; Zaumseil, J.; Chang, J.-F.; Ou, E.C.-W.; Ho, P.K.-H.; Sirringhaus, H.; Friend, R.H. General observation of n-type field-effect behaviour in organic semiconductors. Nature 2005, 434, 194.
  4. Weng, J.; Chen, W.; Xia, W.; Zhang, J.; Jiang, Y.; Zhu, G. Low-temperature solution-based fabrication of high-k HfO2 dielectric thin films via combustion process. J. Sol-Gel Sci.Technol 2017, 81, 662-668.
  5. NIST X-ray Photoelectron Spectroscopy Database online http:// srdata.nist.gov/xps/Default.aspx
  6. The authors corrected Figure 6b. However, nobody shall understand how and why the red straight lines are drawn as shown. And an additional comment; it is highly encouraged to plot the source-gate current in Figure 8a-c to show the insulating properties of the dielectric layers.

Response: Thanks to the reviewer comment. Following the Reviewer comment, the figure 6b had been updated. The line is drawn in figure 6b is the tangent line show us the band gap of the die-electric layer of different annealing temperature. Thanks to the reviewer for the additional comment about the plot of the source-gate current of the Figure 8a-c. In our current experiment, the gate dielectric layer is SiO2, which is purchased from commercial companies and show high-quality insulation characteristics. In our group, we compared the I-V characteristics of as-purchased SiO2 and ALD-driven Al2O3 gate dielectrics, as shown in following figure. It can be noted that the insulating properties is excellent for SiO2 layer. Thank the reviewer for good suggestion, in previous device measurements, we ignored the measurement of source-gate current. In future, we will follow your constructive suggestion, the measurement of source-gate current should be considered in the investigation of TFT devices.

Figure. (a) C-f and (b) I-V of the thermally grown SiO2 and ALD-derived Al2O3 dielectric

Round 3

Reviewer 2 Report

The authors addressed my concerns to revise the manuscript. The current manuscript can be accepted for publication in nanomaterials.